# Objective Assessment of Sleep Patterns among Night-Shift Workers: A Scoping Review

**DOI:** 10.3390/ijerph182413236

**Published:** 2021-12-15

**Authors:** Seunghwa Shin, Su-Hyun Kim, Bomin Jeon

**Affiliations:** 1Department of Nursing, Andong Science College, Kyungpook, Andong 36616, Korea; sswha@hanmail.net; 2College of Nursing, Kyungpook National University, Daegu 41944, Korea; okddeolme12@hanmail.net; 3Research Institute of Nursing Science, College of Nursing, Kyungpook National University, Gukchaebosang-ro 680, Daegu 41944, Korea

**Keywords:** shift work schedule, sleep, sleep disorders, circadian rhythm, wearable electronic devices

## Abstract

In this scoping review of the literature, we identified the types and the parameters of objective measurements to assess sleep patterns among night-shift workers. We conducted a literature search using electronic databases for studies published from 1991 to 2020 and charted and summarized key information. We included 32 studies in the review. Polysomnography was used in 6 studies and wearable sleep detection devices were utilized in 26 studies. The duration of sleep assessment using the wearable devices ranged from 1 day to ≥4 weeks, and more than half of the studies collected data for >2 weeks. The majority of the studies used subjective questionnaires, such as the Karolinska Sleepiness Scale, Epworth Sleepiness Scale, and Pittsburgh Sleep Quality Index, in addition to objective sleep measurements. Total sleep time was the most common parameter, followed by sleep efficiency, sleep onset latency, and time or frequency of being awake. As the utilization of wearable devices to assess the sleep patterns of night-shift workers is expected to increase, further evaluation of device accuracy and precision, optimal data collection period, and key parameters is warranted.

## 1. Introduction

Shift work has been recognized as having a negative influence on health and safety [1]. Working on the night shift, usually from 11 p.m. to 7 a.m. or 12 midnight to 8 a.m. [2], typically causes changes in the sleep–wake patterns, which can lead to transient periods of misalignment between circadian rhythms that are detrimental to health [3]. This includes effects on sleep patterns. Furthermore, sleeping at times that are not optimal may also affect health, with optimal timings being determined by circadian rhythms. With repeated night shifts, sleep loss accumulation and chronic sleep deprivation can occur [4]. The literature suggests that working on the night shift is associated with increased risks of health problems, such as cardiovascular and metabolic diseases, depression, reproductive problems, and cancer [2].

The assessment of sleep patterns among night-shift workers is an essential step in identifying individuals with sleep disturbance and conducting interventions for them [4]. The tools frequently used to measure sleep patterns among night-shift workers are self-report questionnaires (e.g., Pittsburg Sleep Quality Index); however, these can have some drawbacks in assessing night-shift workers’ sleep patterns that might change day by day according to their shift work schedule [3]. Moreover, the data obtained by self-report questionnaires are often confounded with the reporter’s emotional and physical status, which can lead to the underestimation or overestimation of sleep patterns [5]. Therefore, an objective assessment of sleep patterns is important for assessing the sleep disturbances that stem from night-shift work.

Recent advancements in sleep detection technologies have led to the development of measurement tools that involve the use of sensing or signal equipment and recording or data processing [4]. Polysomnography (PSG) has been the “gold standard” for sleep measurement to date, but many wearable sleep detection devices can assess sleep parameters derived from the quantifications of physical activity, heart rate, and electrodermal activity [6].

Given the benefits of collecting objective sleep data, it is advisable that such tools are utilized to assess the sleep patterns of night-shift workers. However, little research exists on this topic. Therefore, in this study, we conducted a scoping review of the literature on the types and the parameters of the objective measurements used to assess sleep patterns among night-shift workers. We conducted this review using the framework of Arksey and O’Malley [7]. This study aimed to review several types of objective assessment tools for measuring the sleep patterns of night-shift workers.

## 2. Methods

### 2.1. Identifying Studies

We conducted our literature search using 11 databases, namely PubMed, Cochrane Library, Web of Science, CINAHL, Scopus, DBpia, KISS, Kmbase, NDSL, RISS, and Google Scholar, for studies published from January 1991 to December 2020 (Figure 1). Keywords used for the database search included: “shift work”, “night work” and “rotating work”; “sleep”, “sleep quality”, “insomnia”, “sleep disturbance”, “sleep deprivation”, “sleep problem” and “sleep disorder”; and “assessment”, “measurement”, “tool”, “device”, “wearable device”, “Actigraph”, “Actiware”, “sensor”, “smart”, “mobile” and “band”. We used the Boolean search strategy to find the studies in the databases (Appendix A).

### 2.2. Study Selection

The inclusion criteria were studies involving night-shift workers, demonstrating quantitative sleep data obtained using devices, and having been published in peer-reviewed English- or Korean-language journals. The exclusion criteria were qualitative studies, secondary data analyses, studies not including sleep parameters, and unpublished theses or dissertations. We selected the studies using the Preferred Reporting Items for Systematic Reviews and Meta-Analyses (PRISMA) [8] (Figure 1). All investigators searched and reviewed the literature independently and then reached an agreement through discussions about the final list of studies.

### 2.3. Charting the Data and Summarizing the Results

We charted key information items obtained from the studies, including purpose, methodology, sleep measurement tools, and results [6]. We then summarized the types and the parameters of the sleep measurements (Table 1 and Table 2).

## 3. Results

### 3.1. Study Characteristics

The database search returned 3059 studies. From the title and abstract review, we selected 45 studies after excluding duplicates (n = 1232) and ineligible studies (n = 1782). After reviewing the full text, we selected a final list of 32 studies (Figure 1). Of these, 15 (46.9%) were cross-sectional surveys [9,10,11,12,13,14,15,16,17,18,19,20,21,22,23], 11 (34.4%) were RCTs [24,25,26,27,28,29,30,31,32,33,34], 4 (12.5%) were quasi-experimental studies [35,36,37,38], and 2 (6.3%) were cohort design studies [39,40].

Overall, 2822 adults were included in the final 32 studies. The study population of night-shift workers included health care workers (n = 18, 56.3%) [10,12,13,15,18,21,24,25,27,29,31,32,33,35,36,38,39,40], production workers (n = 6, 18.8%) [11,14,19,20,22,30], transportation workers (n = 3, 9.4%) [17,20,28], mixed types of workers (n = 1, 3.1%) [23] and no specific types of workers (n = 4, 12.5%) [9,16,26,34].

Most studies (n = 24, 75%) [10,11,12,13,15,16,17,18,20,21,23,24,25,27,28,29,30,31,33,34,35,36,39,40] were published after 2011. The study types encompassed intervention evaluation studies (n = 15, 46.9%) [24,25,26,27,28,29,30,31,32,33,34,35,36,37,38], observational studies (n = 16, 50%) [9,10,11,12,14,15,16,17,18,19,20,21,22,23,39,40], and studies of the identification of sleep disorders (n = 1, 3.1%) [13].

### 3.2. Devices Used to Assess Sleep Patterns

Of the 32 studies that used devices to measure the sleep patterns of night-shift workers, 6 (18.8%) studies measured sleep patterns in a laboratory [9,13,14,26,29,32] and 26 (81.2%) performed measurements using a wearable device in daily living conditions [10,11,12,15,16,17,18,19,20,21,22,23,24,25,27,28,30,31,33,34,35,36,37,38,39,40]. None of the studies utilized laboratory tests and wearable sleep trackers together. Among the studies measuring sleep patterns in a laboratory, six (18.8%) [9,13,14,26,29,32] utilized polysomnography (PSG) and one out of the six studies used the multiple sleep latency test and PSG together [26].

Among the 26 studies using wearable devices to measure sleep patterns, many used Actigraphy (n = 10, 31.3%) [15,19,20,22,24,25,27,34,36,39] or actiwatch (n = 10, 31.3%) [10,11,12,16,18,28,31,33,37,38] (Table 1). Other devices included the Fitbit series (n = 3, 9.4%) [21,23,40], Somnowatch (n = 1, 3.1%) [30], ReadiBand (n = 1, 3.1%) [35], and home electroencephalography (n = 1, 3.1%) [17]. The duration of sleep measurement using wearable devices ranged widely from 3 days to > 4 weeks. More than half of the studies using the wearable devices (n = 16, 61.5%) collected data for > 2 weeks [10,12,16,17,18,19,20,22,24,25,28,31,34,36,37,40].

For sleep patterns, eight studies (30.8%) collected data including at least two consecutive night shifts among two- or three-shift workers [10,12,17,18,19,22,37,39], and seven studies (26.9%) measured sleep patterns during general work schedules, including night-shift work [20,24,25,28,31,36,40] (Table 2). Six studies (23.1%) compared the sleep patterns of night-shift workers to non-shift workers [11,15,16,21,23,27], whereas five studies (19.2%) only investigated workers with night-shift schedules [30,33,34,35,38].

### 3.3. Objective Parameters for Sleep Patterns

Six studies were conducted in the laboratory. Of the six studies using PSG [9,13,14,26,29,32], we obtained nine parameters of sleep patterns, including total sleep time (TST; n = 6, 100%), rapid eye movement (n = 6, 100%), slow-wave sleep (n = 4, 66.7%), non-rapid eye movement (n = 4, 66.7%), sleep efficiency (SE; n = 4, 66.7%), sleep onset latency (SOL; n = 3, 42.9%), time or frequency of being awake after sleep onset (WASO; n = 3, 50%), and bed time (BT; n = 1, 16.7%) (Table 1).

From the 26 studies using wearable sleep detection devices [10,11,12,15,16,17,18,19,20,21,22,23,24,25,27,28,30,31,33,34,35,36,37,38,39,40], we obtained 7 parameters of sleep patterns, including TST (n = 24, 92.3%), SE (n = 12, 46.2%), SOL (n = 10, 38.5%), time or frequency of WASO (n = 6, 23.1%), bed time (n = 5, 19.2%), wake time (n = 4, 15.4%), and nap length or time (n = 3, 11.5%) (Table 1). In the 26 studies using the wearable devices, only 2 reported device accuracy and sensitivity [21,27]. Niu et al. reported that Actigraphy had inter- and intra-instrumental reliabilities with acceleration counts and step data from a mechanical shaker plate, such as coefficient of variation (CV) inter = 1.2% and CV intra = 1.1% for steps and CV inter = 3.5% and CV intra = 2.9% for counts [31]. Shin et al. reported that the Fitbit tracker has an accuracy of 96% and a test–retest reliability coefficient of r = 0.92 [21].

### 3.4. Other Additional Measurements

Of the 32 studies that used devices to measure the sleep patterns of night-shift workers, 23 (71.8%) used self-report sleep questionnaires [9,11,12,13,14,15,16,17,18,20,23,24,25,26,28,29,31,32,33,34,38,39,40], 17 (53.1%) used a sleep diary [12,15,16,18,21,22,24,25,26,27,28,29,31,32,34,36,38], and 13 (40.6%) used both [12,15,16,18,24,25,26,28,29,31,32,34,38] in addition to the objective measurement. The most common self-report questionnaire used to assess sleep quality was the Pittsburg Sleep Quality Index (n = 4, 12.5%) [15,17,23,39]. Others included the Visual Analog Scale and Berlin Questionnaire (n = 3, 9.4% for each) [13,16,24], as well as the Sleep Quality Scale, Overran–Snyder–Halpern Sleep Scale, and Subject Sleep Questionnaire (n = 1, 3.1% for each) [9,33,34]. The self-report questionnaires used to measure sleepiness included the Karolinska Sleepiness Scale (KSS; n = 7, 21.9%) [12,18,20,25,26,28,31], Epworth Sleepiness Scale (ESS; n = 5, 15.6%) [11,14,29,33,40], and Stanford Sleepiness Scale (n = 3, 9.4%) [14,32,38].

The other major variables included: vigilance (n = 13, 38.2%) [12,14,16,18,25,26,29,31,32,33,35,37,38]; mood status, such as depression or anxiety (n = 6, 18.8%) [15,21,26,29,34,38]; performance (n = 4, 12.5%) [12,18,32,36]; quality of life (n = 3, 9.4%) [11,15,40]; fatigue (n = 2, 6.3%) [11,14]; and turnover intention, eating, work load, blood pressure and respiratory infection (n = 5, 15.6%) [10,21,24,26,39]. Three studies (9.3%) assessed the biomarkers of saliva or blood melatonin (Table 2) [29,33,37].

## 4. Discussion

In this scoping review, most of the study participants were health care providers working the night shift. Given that shift work is conducted in diverse types of occupations, the investigations about objective sleep quality associated with night-shift work must be expanded to include other types of jobs. Of the 32 studies in this review, 26 utilized wearable sleep detection devices and 6 used PSG to assess sleep patterns. This review demonstrated that, with the advancement of sleep detection technology using sensors, wearable devices have been gaining popularity. These devices have the advantage of being simple, inexpensive, and easy to use in daily life in comparison to the high cost and inconvenience of PSG, which has been the “gold standard” for sleep measurement [41]. The objective assessment of sleep quality using wearable devices would enhance sleep disturbance detection among night-shift workers. By this, they may gain benefits such as the management of sleep disturbances that might be associated with job performance, accidents, and increased risk of health problems [4].

Most of the wearable devices used to measure sleep patterns were worn on the wrist; Actigraphy and Actiwatch were the most common. In some studies, the Fitbit series was used on the nondominant wrist to check sleep patterns. Actigraphy and Actiwatch, which include a microelectromechanical systems accelerometer, have shown high consistency in the sleep parameters when compared to PSG [42], indicating high reliability and validity for use in clinical practice. One study has reported that wearable devices based on light sensors, such as the Fitbit HR, have a higher sensitivity than Actigraphy for measuring sleep parameters such as TST, SE, and SOL [41]. New types of wearable devices, such as glasses, watches, and bands, offer a wide range of user choices, and also provide real-time data by synchronizing the device with a mobile phone or personal computer [43]. Owing to the development of information technology, various forms of wearable devices are being developed and their use is expected to increase in the future because they are inexpensive, convenient to access, and can be worn for a long period regardless of time or place.

However, in this review, only two studies reported the sensitivity, accuracy, or specificity of the wearable devices for sleep assessment. Most devices are classified as “wellness” products, except for Actigraphy, and have not yet obtained approval from the United States Food and Drug Administration [44]. The findings suggest the need for standardization through various repeated studies and clinical use, so that the sleep parameters measured by wearable devices can be validated [43]. Additionally, the sensitivity, specificity, or accuracy of wearable devices have only been tested in participants who were healthy or had insomnia or other sleep disorders [41,45,46]; only a few studies have demonstrated the sensitivity, specificity, or accuracy of these devices in night-shift workers. Considering the high prevalence of sleep disorders among these workers [3], the investigation of the validity and usefulness of wearable sleep detection devices is needed.

A previous study reported that the main clinical characteristics of people with sleep disturbance from night-shift work are shorter sleep duration and sleepiness [4]. In this review, we categorized 11 objective sleep parameters, including TST, SE, SOL, and WASO. Among them, TST was the most common parameter. TST, as measured by a wearable device, was minimally different from the values assessed using PSG in adults with insomnia. Since sleep duration was demonstrated to be a mediator of metabolic syndrome among female hospital employees working in an alternating day- and night-shift work schedule [47], TST can be used as one of the reliable indicators of sleep quality in night-shift workers [42,45]. Moreover, a previous study suggested the inclusion of sleep duration, sleep and wake times, time required to fall asleep, and number and duration of awakenings during the sleep episode to assess sleep quality in shift workers. It is believed that these sleep quality parameters could be efficiently assessed using wearable devices [4].

In addition to the objective measures of sleep, self-report questionnaires are often used to assess the subjective sleep quality or degree of sleepiness that can occur in a specific situation. Most of the studies used KSS, which evaluates sleepiness at a specific time of the day [48], and ESS, which evaluates the degree of sleepiness in a specific situation [49]. Sleepiness is a common complaint among shift workers and can influence job performance or cause accidents [4]; thus, KSS and ESS are useful measures for assessing the consequences of shift work. Sleep disturbance among shift workers is characterized by insomnia or excessive sleepiness that is associated with a recurring work schedule [4]. Thus, the combined use of self-report questionnaires regarding subjective sleep quality or sleepiness in addition to objective sleep measurement would be desirable to assess the sleep patterns of shift workers.

The duration of sleep pattern assessment in night-shift workers was diverse, from 1 day to >4 weeks. To identify the degree and pattern of sleep disturbance caused by shift work, studies have suggested assessing a minimum of 1 week and ideally >2 weeks of continuous recordings of objective measurement by a sleep detection device along with a sleep diary [4]. Additionally, the assessment should include episodes of both the shift work schedule and days off [49]. Overall, 16 studies collected data on sleep patterns for >2 weeks [10,12,16,17,18,19,20,22,24,25,28,31,34,36,37,40], reporting changes in sleep patterns that could appear in accordance with changes in the work schedule of two- or three-shift workers. In this review, the majority of the studies used a sleep diary in addition to objective measurement, which has been reported to be useful for documenting the factors that can affect sleep over time [42].

This scoping review has a few limitations. First, the studies retrieved for the analysis were limited to English- and Korean-language publications. Second, we have focused on the objective measurements obtained from devices but did not include biophysiological measurements associated with sleep, such as core body temperature, melatonin, and cortisol [50].

## 5. Conclusions

With the recent advancements in sleep detection technology that uses sensors, the use of wearable devices in daily living conditions has been gaining popularity for measuring sleep. Wearable devices, which are convenient and reliable, can be used to assess the need for and progress of interventions for sleep disorders in night-shift workers, who may experience changes in their sleep patterns due to disrupted circadian rhythms. The key objective parameters for assessing sleep quality among night-shift workers, such as sleep duration, the timing of sleep and wake times, time required to fall asleep, and number and duration of awakening during the sleep episode, can be efficiently measured using wearable sleep devices. To investigate the consequences of shift work, it is desirable to assess the sleep patterns of night-shift workers for >2 weeks with the combined use of a sleep diary and subjective questionnaire. More studies are needed to test the validity of and identify optimal duration and key parameters for wearable sleep detection devices.

## Figures and Tables

**Figure 1 ijerph-18-13236-f001:**
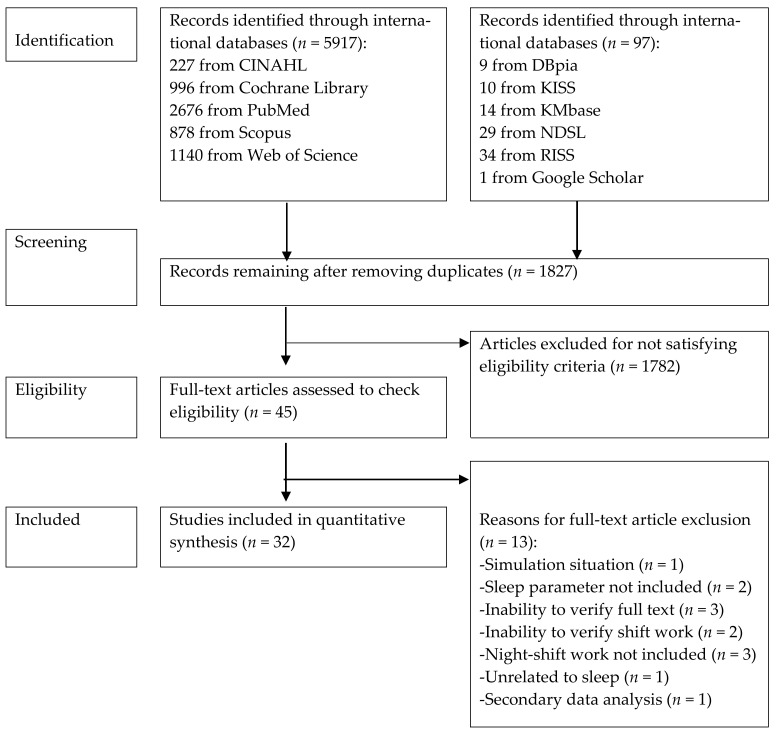
PRISMA flow diagram.

**Table 1 ijerph-18-13236-t001:** Devices and sleep parameters used in the included studies (*N* = 32).

Studies	Device			Sleep Parameter									
	Polysomnography	Multiple Sleep Latency Test	Wearable Sleep Trackers	Bedtime	Nap Length	NREM Sleep	REM Sleep	Sleep Efficiency	Sleep Onset Latency	Slow-Wave Sleep	Total Sleep Time	Awake after Sleep Onset	Wake Time
*Studies conducted in the laboratory*													
Geiger-Brown et al., 2013 [9]	√						√	√		√	√		
Rahman et al., 2013 [10]	√					√	√	√	√	√	√	√	
Czeisler et al., 2009 [11]	√	√				√	√	√	√		√	√	
Hossain et al., 2004 [12]	√					√	√	√	√	√	√	√	
Smith Coggins et al., 1997 [13]	√						√				√		
Akerstedt et al., 1991 [14]	√			√		√	√			√	√		
*Studies utilizing wearable devices*													
Chen et al., 2020 [15]			√								√		
James et al., 2020 [16]			√					√	√		√		
Lavigne et al., 2020 [17]			√					√			√		
Loef et al., 2020 [18]			√								√		
Mentink et al., 2020 [19]			√								√	√	
Shin et al., 2020 [20]			√					√					
Thottakam et al., 2020 [21]			√						√		√	√	
Zeitzer et al., 2020 [22]			√								√		
Barger et al., 2019 [23]			√								√		
Basner et al., 2019 [24]			√								√		
Loew et al., 2019 [25]			√								√		
Mulhall et al., 2019 [26]			√	√				√	√		√	√	√
Low et al., 2018 [27]			√					√	√		√		
Pylkkönen et al., 2018 [28]			√								√		
Kwak et al., 2017 [29]			√					√	√		√	√	
Kwon et al., 2017 [30]			√								√		
Niu et al., 2017 [31]			√					√	√		√	√	
Sallinen et al., 2017 [32]			√								√		
Fernandes-Junior et al., 2016 [33]			√					√	√		√		
Sadeghniiat-Haghighi et al., 2016 [34]			√					√	√		√	√	
Shea et al., 2014 [35]			√		√								
Ftouni et al., 2013 [36]			√	√				√			√		√
Sasseville et al., 2010 [37]			√	√				√	√		√		
Moon et al., 2004 [38]			√		√						√		
Ko et al., 2002 [39]			√	√				√	√		√		√
Park et al., 2000 [40]			√	√	√						√		√

**Table 2 ijerph-18-13236-t002:** Descriptive summary of the included studies (N = 32).

Author, Year, Country	Purpose	Design and Population	Outcome	Results
*Studies conducted in the laboratory*				
Geiger-Brown et al., 2013; USA [9]	To describe the prevalence of breathing symptoms in nurses with sleep disorders and examine the validity of BQ to screen for sleep apnea in the sleep-deprived group.	Cross-sectional study Health care workers (*N* = 21)	Sleep	BQ produced valuable data regarding sleep apnea symptoms.
Rahman et al., 2013; Canada [10]	To examine the effects of filtering short wavelengths during night shifts on sleep and performance.	RCT crossover Health care workers (*N* = 9)	Sleep Mood Vigilance Melatonin (saliva)	Filtering short wavelengths can reduce sleep disruption and improve performance in rotating-shift workers.
Czeisler et al., 2009; USA [11]	To assess the effects of armodafinil on the physiological propensity for sleep and cognitive performance during night-shift hours for workers with chronic shift work disorder.	RCT Shift workers (*N* =245)	Sleep Vigilance Mood Others	Armodafinil improved wakefulness during scheduled night work, which raised the mean nighttime sleep latency above the level that indicates severe sleepiness during the daytime.
Hossain et al., 2004; Canada [12]	To evaluate short- and long-term impacts of shift schedule changes on sleep and performance.	Cross-sectional study Production workers (*N* = 241)	Sleep Vigilance Fatigue	Improved subjective and objective measures of sleep and performance on a new 10-h night-shift schedule.
Smith Coggins et al., 1997; USA [13]	To evaluate the effectiveness of broad, literature-based, night-shift work intervention for the enhancement of emergency physicians’ adaptation to night rotations.	RCT Health care workers (*N* = 6)	Sleep Vigilance Performance	Rotating shift work schedules improved physicians’ sleep, performance, and mood on the night shift.
Akerstedt et al., 1991; Sweden [14]	Sleep PSG in shift workers, in which recordings were made at 2-year intervals.	Cross-sectional study Shift workers (*N* = 20)	Sleep	Core variables of sleep showed considerable interindividual stability across time; 2-year exposure to rotating-shift work did not affect sleep in experienced shift workers.
*Studies utilizing wearable devices*				
Chen et al., 2020; USA [15] ^†^	To assess the eating patterns, sleep, and physical activity of health care workers on three different shifts.	Cross-sectional study Health care workers (*N* = 14)	Sleep Other	Shift work was associated with increased calorie intake, high-fat and -carbohydrate diets, and sleep deprivation.
James et al., 2020; USA [16] *	To physiologically measure the sleep patterns and predict cognitive decline of nurses working both 12-h day and night shifts to address the concern about sleep restriction among health care workers.	Quasi-experimental study Health care workers (*N* = 90)	Sleep Vigilance	Differences were observed in sleep quantity, efficiency, and latency based on shift type and duty.
Lavigne et al., 2020; Canada [17] **	To assess changes in the sleep and vigilance of underground miners during long periods of extended shifts.	Cross-sectional study Shift workers (*N* = 70)	Sleep Vigilance	Underground miners exhibited good sleep quality despite evidence of limited circadian adaptation in nighttime vigilance.
Loef et al., 2020; Netherlands [18] ^†^	To examine the mediating roles of sleep, physical activity, and diet between shift work and respiratory infections.	Cohort study Health care workers (*N* = 396)	Sleep Others	Shift workers had a higher incidence of ILI/ARI, which was partly mediated by poor sleep quality.
Mentink et al., 2020; Netherlands [19] ^†^	To explore sleep disruption: Increased efficiency in generating deep sleep during work weeks and rebound sleep during rest weeks.	Cross-sectional study Transportation workers (*N* = 10)	Sleep	Increased efficiency in generating deep sleep during work weeks was more likely to be a compensatory mechanism for sleep disruption in the maritime pilot cohort than rebound sleep during rest weeks.
Shin et al., 2020; South Korea [20] **	To identify the influence of night-shift work and SE on fatigue, depression, and turnover intention among hospital nurses.	Cross-sectional study Health care workers (*N* = 64)	Sleep Mood Others	Nurses working night shifts demonstrated lower SE and higher levels of fatigue and turnover intention than non-shift-working nurses.
Thottakam et al., 2020; UK [21] *	To investigate the feasibility and acceptability of melatonin administration in night-shift workers and its effects on sleep measures and attention/concentration tasks.	RCT crossover Health care workers (*N* = 25)	Sleep Vigilance Melatonin (blood)	Double-digit addition, a concentration/attention task, improved with melatonin treatment.
Zeitzer et al., 2020; USA [22] *	To determine if a hypocretin receptor antagonist would enable shift workers to have more daytime sleep.	RCT Shift workers (*N* = 19)	Sleep Mood	Suvorexant group increased their objective and subjective TST.
Barger et al., 2019; USA [23] ^‡^	To compare work hours and sleep in resident physicians on extended-duration work rosters to extended-duration shifts or rapidly cycling work rosters.	RCT crossover Health care workers (*N* = 362)	Sleep Workload	Residents on rapidly cycling work rosters had improved sleep duration.
Basner et al., 2019; USA [24] ^‡^	To establish the sleep and alertness among interns in flexible programs compared to those in standard programs.	National-cluster RCT Health care workers (*N* = 398)	Sleep Vigilance	There was no more chronic sleep loss or sleepiness across trial days among interns in flexible programs compared to those in standard programs.
Loew et al., 2019; USA [25] ^‡^	To examine Plan–Do–Study–Act QI model to improve pediatric medical provider sleep and communication during night shifts.	Quasi-experimental study Health care workers (*N* =49)	Sleep Performance	Provider-based standardization of paging communication was associated with changes in medical-specific communication between nurses and providers.
Mulhall et al., 2019; Australia [26] ^†^	To investigate the objective and subjective sleepiness and driving events during short work.	Cross-sectional study Health care workers (*N* = 33)	Sleep Vigilance Performance	Subjective and objective sleepiness and driving events increased following night shifts, even during short commutes, and were exacerbated by interactions between the circadian phase and duration of wakefulness.
Low et al., 2018; Singapore [27] ^‡^	To examine and compare the activity levels, sleep, fatigue, and professional QoL between residents working on night floats and those on overnight calls.	Cohort study Health care workers (*N* = 49)	Sleep QoL	Physical activity and amount of sleep were not significantly different between night-float and on-call residents.
Pylkkönen et al., 2018; Finland [28] ^‡^	To examine the effects of educational intervention on long-haul truck drivers’ sleepiness while driving and the amount of sleep between work shifts.	RCT with repeat measures Transportation workers (*N* = 49)	Sleep	No significant intervention-related improvements occurred in driver sleepiness, prior sleep, or SCM use while working on night and early morning shifts.
Kwak et al., 2017; South Korea [29] **	To investigate the sleep patterns of shift-working and daytime psychiatric nurses using objective and subjective assessments for sleep.	Cross-sectional study Health care workers (*N* =48)	Sleep Mood QoL	Shift-working nurses experienced more sleep disturbances in subjective and objective aspects of sleep than daytime-working nurses.
Kwon et al., 2017; South Korea [30] **	To verify the relationship between physical activities and sleep characteristics of shift workers to improve their health problems.	Cross-sectional study Mixed-shift workers (*N* = 53)	Sleep Others	Shift workers showed an imbalance between physical activity and sleep due to work schedules and sleep duration.
Niu et al., 2017; Taiwan [31] **	To explore the differences in sleep parameters between nurses working on slow, forward-rotating shifts and those working on fixed day shifts.	RCT Health care workers (*N* =62)	Sleep	TST in nurses working on evening-rotating shifts was higher than in nurses working on day-rotating, night-rotating, or fixed day shifts.
Sallinen et al., 2017; Finland [32] ^‡^	To measure the pilots’ sleep–wake patterns and on-duty alertness levels and management strategies.	Cross-sectional study Transportation workers (*N* = 86)	Sleep	Short- and long-haul flight duty periods covering the whole domicile night were most consistently associated with reduced sleep–wake ratio and subjective alertness.
Fernandes-Junior et al., 2016; Brazil [33] **	To evaluate the sleep time, fatigue, and QoL of night-shift workers and verify the relationship between these variables with the presence or absence of children in different age groups.	Cross-sectional study Production workers (*N* = 78)	Sleep Fatigue QoL	Shift workers without children had higher sleep time during working days and were less likely to experience fatigue during night work than workers with children, regardless of children’s ages.
Sadeghniiat-Haghighi et al., 2016; Iran [34]	To evaluate the efficacy of 3-mg melatonin taken 30 min before nighttime sleep on shift workers with difficulty falling asleep.	RCT crossover Production workers (*N* = 39)	Sleep	Melatonin treatment increased SE and decreased SOL. Effects of melatonin on TST and WASO were not significant.
Shea et al., 2014; USA [35] *	To evaluate the intern and patient outcomes associated with protected 3-h nocturnal nap periods.	RCT Health care workers^‡^ (*N* = 179)	Sleep Vigilance	Protected 3-h sleep periods resulted in more sleep during call and reductions in periods of prolonged wakefulness that provide a plausible alternative to 16-h shifts.
Ftouni et al., 2013; Australia [36] ^†^	To assess the relationships between the sleepiness and incidence of adverse driving events in nurses commuting to and from night and rotating shifts.	Cross-sectional study Health care workers (*N* =27)	Sleep Vigilance Performance	For the shift-working group, self-reported sleepiness, drowsiness, and driving events were higher during commutes following night shifts compared to commutes before night shifts.
Sasseville et al., 2010; Canada [37] ^†^	To investigate the possibility of adaptation in shift workers who are exposed to blue–green light at night, combined with blue blockers during the day.	Quasi-experimental study Production workers (*N* = 4)	Sleep Vigilance Melatonin (saliva)	Strategic exposure to short wavelengths at night and daytime using blue-blocker glasses improved sleep, vigilance, and performance.
Moon et al., 2004; South Korea [38] ^†^	To investigate the effects of sleep–wake behavior for shift workers on a continuous, full-day, 3-shift system of backward rotation.	Cross-sectional study Production workers (*N* = 59)	Sleep	Sleep length at home during night shifts decreased compared to morning or evening shifts. Night-shift NL increased in all sections compared to morning or evening shifts.
Ko et al., 2002; South Korea [39] *	To evaluate the effects of bright light on adaptation to night-shift work.	Quasi-experimental study Health care workers (*N* = 5)	Sleep Vigilance Mood	Subjective feelings, attention, and alertness were enhanced during light exposure.
Park et al., 2000; South Korea [40] ^†^	To investigate the sleep–wake behavior and effects of aging on the tolerance of night shifts in the continuous, full-day, 3-team, 3-shift system.	Cross-sectional study Production workers (*N* = 12)	Sleep	For those on night duty, TST decreased, the number of naps and NL during on-duty or off-duty periods increased, and level of activity decreased with increasing age.

*, sleep checked only on night-shift work schedule; **, separate for night-shift and non-shift workers; †, sleep checked, including regular consecutive night-shift workers; ‡, sleep checked for general shift schedule; RCT, randomized controlled trial.

## Data Availability

The data presented in this study are available on request from the corresponding author.

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
