# Peer review of "Objective Assessment of Sleep Patterns among Night-Shift Workers: A Scoping Review"

_ijerph, 2021, doi:10.3390/ijerph182413236_

Round 1

Reviewer 1 Report

The introduction should be more detailed, focusing on the relevance and impacts of sleep disturbs/quality on workers’ health. Only PSQI is cited as questionnaire, there are many others.

Line 28: it would be useful to provide a definition of night shift work.

I do not understand why “sleep quality” was not used in literature search if “The goals of this study were to obtain the best guidance for selecting objective assessment tools to measure night-shift workers’ sleep quality…”

The footnote in table 1 does not provide all the acronyms; moreover, the table is very chaotic. Authors should consider to provide a more readable one.

The discussion is not easy to read. Authors should focus more on take home messages about the objective assessment of sleep quality and the repercussion on workers’ health.  

Author Response

I have submitted the file.

Reviewer 2 Report

To the authors,

In my opinion, a scoping review on this topic could be very helpful and save many other researchers a lot of time. However, I consider the synthesis of the review material to be poor and the conclusions contradictory and tenuous. I elaborate on these along with more minor limitations in the following specific points:

  • Lines 28-30: “Individuals who work at night…have a greater risk for [various diseases]....”

Although I expect this to be true, it is too definitive. Associations have been made but this does not confer causal risk. For instance, the leading authority on cancer research (IARC) concluded night shift work is "probably" carcinogenic in 2019. They did not extend classification to “definitely”.

  • Line 30: What is reference number 3 citing in this sentence?

  • Line 30-31: “Due to misalignment between work shifts and circadian rhythms, working the night shift involves changes in the sleep-wake pattern, which can lead to sleep disturbance”.

I can appreciate the point you are trying to make here but the sentence is poorly formulated. For instance, what is misalignment between work shift and circadian rhythms? And why is it only due to this misalignment that working the night shift involves changes in the sleep-wake pattern? I would suggest a reformulation similar to the following: 'Working the night shift typically involves changes in the sleep-wake pattern. Changes in the sleep-wake pattern can lead to transient periods of misalignment between circadian rhythms which can be detrimental to health. This includes affecting sleep quality. Furthermore, having to sleep at times that are not optimal-with optimal timing being determined by circadian rhythms-may also affect health.  

  • Lines 35-36: “…the occurrence of any negative effects.”

What negative effects? Do you mean CVD, depression, reproductive problems, etc? If so, in the first paragraph you link night work to these and night work to sleep disturbance but you do not provide a link from sleep to these negative effects. If not these effects, then what effects? Why is it essential to identify and intervene with individuals at risk for sleep disturbance? Does "at risk" mean sleep disturbance will defintiely occur? Does sleep “quality” assessment provide evidence that sleep “disturbance” will or will not occur?

  • Lines 37-38: “self-report questionnaires...can have some drawbacks when used with shift workers who sleep during the day…”

What drawbacks are you referring to regarding use with shift workers who sleep during the day?

  • Lines 56-57: “The goals of this study were to obtain the best guidance for selecting objective assessment tools”.

How these goals are to be reached or and how to evaluate if the goals had been reached is not discussed in the methods and these goals are not referred to in results or discussion.

  • Line 65 & Appendix A: Not a limitation but perhaps of interest, if you include “sleep” per se as a search term in PubMed or WOS then there is no need to also include “sleep disturbance”, “sleep problems,” “sleep + another word”, etc. Hits from these combined terms will all be returned by using “sleep” per se.

  • Line 123: “44 studies” and “n=1,232”

The numbers do not fit with what is in your PRISM flow diagram.

  • Lines 134-136: Why provide quality appraisal of the studies if it is of no benefit to the “goals” of the article? If it has no value, it is at best an annoyance and at worst a distraction to the reader. If there is relevance to the quality appraisal, that you wish to elaborate upon, then please do so but also include how you reached the “quality scores” for each study (e.g., what parts of the studies scored better or worse and why do you consider this so?).

  • Line 143: “1(2.9%) used the Multiple Sleep Latency Test…”

Table 1 indicates that 3 studies used the MSLT.

  • Lines 152-156: This information is very difficult to glean from Appendix B. A second table would be preferred rather than using small icons to indicate this information in an already full table.

  • Line 164-165: You describe the MSLT used as both a method to assess- and a parameter of- sleep quality. What is the distinction?

  • Lines 187-192: This does not belong to a section titled “Parameters for Sleep Quality”. Also, why have you included this information?

  • Lines 212-213: “The use of wearable devices based on light sensors is expected to increase in the future.”

Why?

  • Lines 217-219: “Most devices are not approved by the U.S. Food and Drug Administration, suggesting the need for large-scale studies to verify the validity of sleep measurements [48].”

Please explain the link here.

  • Lines 225-231: This paragraph is not a sufficient argument that TST can be considered a reliable indicator of sleep quality for night-shift workers. (1) One single study reporting it as main clinical characteristic is insufficient. (2) That it is the most commonly used is insufficient. (3) Reconciliation with PSG measurments is insufficient. The conclusion also contradicts your call for validity.

  • Lines 232-238: While I accept that if this information was missing, many readers would ask about it, this paragraph still adds nothing to goals of this article. The relevance of this paragraph should be described.

  • Lines 252-253: Can you give some examples of the "in vivo biophysiological measurments" that can be considered as objective assessment of sleep quality among night shift workers to which you are referring? What are in vitro measurments of sleep quality in night shift workers?

  • Conclusions Section: see point 15 above. In addition, the importance of circadian rhythms is mentioned in the first paragraph. Note that sleep time does not necessarily always correspond with the optimum circadian time for sleep (i.e., sleep can occur when rhythms that would typically be in a trough during regular sleep are actually high). See 3rd point above.

  • Table 1: If there are no relevant sleep parameters studied by Gumenyuk et al (2014) or Howard et al (2014), why are they included? What does BT stand for? The layout of the abbreviated devices and parameters makes it hard to distinguish what letters belong to what parameter.

  • There is no discussion as to the importance or relevance of the different sleep parameters. If you are to indicate that TST is a key parameter, then a discussion on the relevance of these parameters should be provided.

  • Discussion of what precisely the devices measure and how this can be translated into measures of sleep quality is missing.

Author Response

I have submitted the file.

Reviewer 3 Report

I reviewed the manuscript ID ijerph-1384223 in which the authors conducted a review of the scientific literature trying to select objective assessment tools to evaluate night-shift workers' sleep quality. I think that the article is interesting and the topic has the potential to be interesting and useful from a scientific point of view.

Indeed, I have only few suggestions. The literature, in my opinion, has not been thoroughly researched and referenced because missing 2021 papers (for example, DOI:10.3389/fphys.2021.628231; DOI:10.3390/ijerph18168378).

Pay attention to the table 1 and appendix B. The descriptions of the table 1 and appendix B are not complete in terms of abbreviations and please, in both, report the abbreviations following the presentation order.

Author Response

I have submitted the file.

Round 2

Reviewer 1 Report

No more issues.